# Factors Contributing to the Comprehensive Use of Food Labels in Jordan

**DOI:** 10.3390/nu15234893

**Published:** 2023-11-23

**Authors:** Amjad Rashaideh, Mohammed Al-Nusair, Ahmad Ali Alrawashdeh, Taha Rababah, Adi Khassawneh, Yazan Akkam, Ayoub Al Jawaldeh, Jomana W. Alsulaiman, Khalid A. Kheirallah

**Affiliations:** 1Department of Public Health, Family Medicine, and Community Medicine, Faculty of Medicine, Jordan University of Science and Technology, Irbid 22110, Jordan; amjad.rashaideh@jfda.jo (A.R.); ohkhasawneh@just.edu.jo (A.K.); 2Faculty of Medicine, Jordan University of Science and Technology, Irbid 22110, Jordan; mohammed.alnusair99@gmail.com; 3Department of Allied Medical Sciences, Faculty of Applied Medical Sciences, Jordan University of Science and Technology, Irbid 22110, Jordan; aaalrawashdeh@just.edu.jo; 4Department of Nutrition and Food Technology, Jordan University of Science and Technology, P.O. Box 3030, Irbid 22110, Jordan; trababah@just.edu.jo; 5Department of Medicinal Chemistry and Pharmacognosy, Faculty of Pharmacy, Yarmouk University, Irbid 21163, Jordan; yazan.a@yu.edu.jo; 6Regional Office for the Eastern Mediterranean, World Health Organization, Cairo 7608, Egypt; aljawaldeha@who.int; 7Department of Pediatrics, Faculty of Medicine, Yarmouk University, Irbid 21163, Jordan; jomana.a@yu.edu.jo

**Keywords:** food label, nutrition label, Jordan, non-communicable disease, nutrition, diet, East Mediterranean Region

## Abstract

Food labels are low-cost, informational tools that can help curb the spread of diet-related non-communicable diseases. This study described consumers’ knowledge, attitudes, and practices related to food labels in Jordan and explored the relationship between knowledge and attitude with comprehensive use of food labels. A cross-sectional, online survey assessed Jordanian adult consumers’ ability to comprehend the nutritional contents of food labels (knowledge score), their attitudes towards food labels (attitude scale), and how frequently they used different parts of food labels (practice scale). Multivariate logistic regression models assessed predictors of comprehensive use of food labels. A total of 939 adults participated in the study. Total mean scores for the practice scale (14 questions), attitude scale (8 questions), and knowledge score (4 questions) were 49.50 (SD, 11.36; min, 5; max, 70), 29.70 (SD, 5.23; min, 5; max, 40), and 1.39 (SD, 1.33; min, 0; max, 4), respectively. Comprehensive users of food labels (26.4%) were more likely female, responsible for grocery shopping, and had higher mean knowledge and attitude scores. Jordanian consumers seem to have good practices and attitudes related to food label use but suboptimal knowledge regarding content. Future interventions should focus more on enhancing knowledge and awareness related to food labels.

## 1. Introduction

The Eastern Mediterranean Region (EMR) has a high burden of non-communicable diseases (NCDs), where the probability of dying from NCDs (22%) is higher than the global rate (18.3%) [1]. In Jordan, a country in the EMR, NCDs are responsible for the great majority of deaths (75.6%), with cardiovascular disease accounting for 34.7% and diabetes accounting for 6.7% [2,3]. With such a burden, the growing impact of obesity and metabolic syndrome on public health necessitates a need to support consumers’ healthier lifestyle choices, especially healthier diets [4]. An effective population-based approach targeting diet and nutrition may then be a critical national need.

Food labeling, a low-cost informational tool, can provide consumers with information on packaged food items, including serving size and nutritional facts [5]. Food labels were helpful in guiding consumers to healthier dietary options by providing information at the point of sale [4]. Yet, others reported food labels to be challenging due to deficient knowledge, negative attitudes, and poor practice [6,7]. Food label use was also related to perceived benefits from food labels as well as confidence in understanding them [8]. A systematic review reported that consumers are often confused by the added information that exceeds the questions they originally had in mind [9]. Food label design, reading time, and language barriers are also factors that impact the use of food labels [10,11]. Some consumers reported being “doubtful” of the precision and honesty of information provided within food labels [12]. Regardless of the reason, reported difficulties in properly using food labels may translate into restricted future use [13].

Such challenges have been reported in developed countries such as the U.S. [14], Australia [15,16], and Canada [17], as well as developing countries including those in the EMR [11,18]. Yet, certain sociodemographic factors and lifestyle choices have been associated with proper use of food labels, such as female sex, high education and income, older age, healthier dietary habits, and engaging in weight loss activities [4,18]. Food label use was also more frequent among those who had higher levels of perceived benefit from food labels and higher confidence in reading and understanding them [8].

While many developing countries are going through the epidemiologic transition, the shift from communicable diseases to chronic diseases necessitates exploring interventions to combat NCDs. Food labeling seems to be an effective strategy to promote healthier diets when used properly. The WHO recommends the implementation of food labeling as a strategy to control and prevent NCDs [19]. However, further investigation of the factors that increase the use of food labels is required to help improve this tool in preventing and controlling NCDs. This study aimed to describe consumers’ knowledge, attitudes, and practices related to food label use in Jordan and to explore the relationship between knowledge and attitude with the comprehensive use of food labels. A better understanding of the characteristics of consumers who use or do not use food labels will better guide public health interventions in controlling and preventing NCDs.

## 2. Materials and Methods

### 2.1. Study Design

This was a cross-sectional survey (online questionnaire) that assessed consumers’ knowledge, attitudes, and practices related to food labels in Jordan. This study was approved by the Institutional Review Board (IRB) of King Abdallah University Hospital and Jordan University of Science and Technology (2022/148/2).

### 2.2. Participants

The participants of this study were adults (≥18 years) living in Jordan who could read and write, have access to a smartphone, and purchase their food items from conventional supermarkets (defined as stores that offer a wide variety of food and household products, have multiple aisles and departments, including fresh produce, dairy, meat, bakery, and non-perishable items). The list of supermarkets available at the Jordan Chamber of Commerce was utilized to select a random sample of supermarkets from each of Jordan’s 12 governorates. The sample in each governorate included four locations in rural areas and four in urban areas. Within each randomly selected location, a QR code, allowing access to the study questionnaire, was distributed to shoppers.

### 2.3. Study Questionnaire

The questionnaire was constructed to collect data on participants’ background characteristics, knowledge about food label contents, attitude towards information within food labels, how frequently they use different parts of food labels, and their understanding of the nutritional contents of food labels based on an example food label. The questionnaire was adapted from a series of investigations with the same objective and validated by an expert panel. The questionnaire contains 41 questions assessing sociodemographic factors (including age, gender, employment status, educational level, and monthly income), as well as topics related to reading food labels such as importance and comprehensiveness of information, ease of understanding, buyers’ background knowledge, participation in the procurement process, grams and portions of sugar and fats, knowledge about best buy dates and best before dates, preparation and storage methods, portion sizes, product names, claims, oils and fat types, and salt and sugar consumption in the last six months.

The knowledge scale was constructed using four questions testing participants’ ability to comprehend the information presented on an example food label (Figure 1, Table 1). Questions for the knowledge factor had one correct answer out of four answer options, and participants would receive a score of one on each question answered correctly. Factor analyses were used to create two major scales for participants’ practice and participants’ attitudes related to food labels. Questions, scored on a 5-point Likert scale, were collapsed into the practice scale utilizing 14 questions assessing how frequently participants read different aspects of food labels (Table 2) and the attitude scale utilizing eight questions assessing participants’ thoughts on the importance of food labels and the value of the information contained within them (Table 3).

### 2.4. Pilot Study

A pilot study was conducted to obtain feedback from the participants on the clarity and content of the questionnaire. For content validity, the questionnaire was reviewed by a panel of food safety and public health experts from the Jordan Food and Drug Administration. The language and length of questions were adapted to ensure applicability and acceptance. The pilot study was performed by 15 individuals using the designed questionnaire. Necessary edits were conducted before distributing the questionnaire to the study participants.

### 2.5. Data Collection

Participants scanning the QR code would be directed to a Google Forms page providing details about the study objectives, voluntary participation, privacy, and confidentiality of collected data, and asking participants to consent to participate. Eligible participants were then directed to take the survey. Data were collected between March and June 2022.

### 2.6. Statistical Analysis

The minimum sample size needed was estimated at 377 participants to produce a 5% margin of error and 80% power to provide a conservative estimate of food label use among participants. Descriptive statistics, numbers, and percentages, as well as means and standard deviations (SD), were performed to describe and represent data. Bivariate analyses were conducted using the Chi-square test for categorical variables and the student’s *t*-test for continuous variables.

Exploratory factor analyses were applied to collapse the practice and attitude questions (items). Two scales were identified (practice and attitude). Within the practice scale, two additional subscales were identified: “nutritional facts” subscale (covering questions related to primarily nutritional values, ingredients, allergy information, etc.;) and “product information” subscale (primarily encapsulating questions related to product name, country, expiration date, etc.). Together, these two subscales explain 99% of the total variance in the responses. The factors were determined to be very reliable (Cronbach’s alpha of 0.93 for the overall practice scale, 0.83 for the nutritional facts subscale, and 0.92 for the product information subscale). The practice scale was used to categorize participants into “comprehensive users” and “non-comprehensive users” of food labels. Comprehensive users were defined as participants who had a total practice score equal to or greater than the 75th percentile.

Two subscales were also identified within the attitude scale: the “label information” subscale, which primarily concerns the use and characteristics of food labels, and the “nutritional values” subscale, which primarily encapsulates items related to fat and sugar contents. Together, these two subscales explain 57.7% of the total variance in the responses. The factors demonstrated good reliability, with Cronbach’s alpha for the overall attitude scale of 0.81, 0.81 for the nutritional values subscale, and 0.68 for the label information subscale.

Backward selection multivariate logistic regression models were constructed to assess factors predictive of comprehensive use (comprehensive users) of food labels while controlling for possible confounding factors. Adjusted odds ratios (aOR) and 95% confidence intervals (CI) were reported. A *p*-value of less than 0.05 was considered for all statistically significant differences. Stata version 16 was used to analyze data.

## 3. Results

A total of 939 adults participated in the study. Most study samples were males (57.6%), were between 40 and 60 years old (48.1%), had university degrees (67.4%), and had a monthly income between 200 and 500 JDs (41.3%) (Table 4). About 71 percent of participants were responsible for household grocery shopping.

Fourteen questions, which assessed how frequently participants read different aspects of food labels, were used to define the overall practice scale. Table 2 presents the means and SDs of all items of the practice scale. Two practice subscales were also identified: product info and nutritional facts subscales. The overall mean practice score was 3.54 (SD ± 0.81). The mean scores of all items ranged between 2.99 (SD ± 1.30) and 4.16 (SD ± 1.00). The mean total (sum of) score was 49.50 (SD ± 11.36), and the median was 50.0 (IQR:42.0–57.0). Appendix A include the results of the factor analyses.

Eight questions assessing participants’ attitudes on the importance of food labels and the value of the information contained within them were used to produce the attitude scale. Table 3 presents the means and SDs of all items of the attitude factor. The mean score was 3.70 (SD ± 0.65). The mean scores of the items ranged between 3.26 (SD ± 1.12) and 4.49 (SD ± 1.68). The mean total (sum) score was 29.70 (SD ± 5.23), and the median was 30.0 (IQR:26.0–33.0).

Participants were asked which nutrients were most important for them to read about on food labels. Calories (48.6%) and fat (47.6%) were the two most important nutrients (Figure 2).

Four questions testing participants’ ability to comprehend the information presented on an example food label (Figure 1) were used to produce the knowledge score. Table 1 presents all items of the knowledge scale with the number (%) of participants that answered each question correctly. The overall mean score was 0.35 (SD ± 0.33), and the mean total (sum) score was 1.39 (SD ± 1.33).

A summary of the practice and attitude scales and the knowledge score with minimum, maximum, and mean ± SD values is provided in Table 5.

The practice scale was used to categorize participants into food label “comprehensive users” and “non-comprehensive users”. The number of comprehensive users (total practice score equal to or greater than the 75th percentile [57.0]) was 248 (26.4%). Table 6 shows the characteristics of comprehensive users and non-comprehensive users. Comprehensive users were significantly more frequently female (*p* = 0.011), graduates of higher education (*p* = 0.029), and responsible for shopping for food (*p* < 0.001) compared to their respective counterparts. Comprehensive users also had a more positive attitude toward food labels, thinking more highly of the importance and value of the information contained within them, and had a keener ability to understand that information, indicated by greater mean attitude and knowledge scores, respectively (*p* < 0.001 and *p* = 0.002, respectively).

Table 7 shows the results of a multivariate logistic regression model constructed to identify predictors of comprehensive use of food labels. Females were about twice as likely to be comprehensive food label users compared to males (adjusted odds ratio, 95% confidence interval = 1.75, 1.24–2.48). Participants who reported being responsible for grocery shopping were 2.47 times as likely to be comprehensive users compared to their counterparts (2.47, 1.63–3.74). Higher attitude (1.39, 1.30–1.49) and knowledge (3.81, 1.37–10.57) scores were also significant predictors for comprehensive use of food labels. While age and income were not predictors of comprehensive use, higher education was noted to increase the likelihood of comprehensive use. However, this increase was not statistically significant (1.46, 1.00–2.13).

## 4. Discussion

Given Jordan’s epidemiologic transition, food label use should be properly understood to guide interventions supporting a healthier diet [3]. This study characterized comprehensive users of food labels and identified their relationship to selected socio-demographic variables along with knowledge and attitudes related to food labels. The results outlined the different aspects surrounding consumers’ comprehensive use of food labels and produced scales describing how frequently participants read different aspects of food labels, denoted by the practice scale; consumers’ attitudes on the importance of food labels and the value of the information contained within them, indicated by the attitude scale; and their ability to understand the information contained within a food label, denoted by the knowledge scale. The results indicated that the likelihood of comprehensive use of food labels was higher among female consumers, consumers responsible for grocery shopping, and consumers with more positive attitudes and knowledge related to food labels. Furthermore, while participants seem to have good attitudes and practices regarding food labels, their knowledge seems to be suboptimal. This emphasizes the importance of targeting public awareness and education on food label reading and interpretation in those with low socioeconomic status. Current food labels in Jordan may be ineffective without such educational programs, which leads to wider dietary and health inequalities and higher rates of obesity.

The use of food labels was addressed in relation to the Health Belief Model constructs [20]. It was noted that consumers who may sense a “personal threat” from unsafe food and consider “doing something about it” will be more likely to engage in food safety behavior. Believing that nutrition influences health, being well-educated, and being motivated by health concerns, all increased the likelihood of using nutrition labels [21]. Higher levels of perceived benefit and higher confidence in reading and understanding food labels were associated with higher frequency of food label use [22]. Similarly, the results suggest that comprehensive users had a more positive attitude toward food labels, thinking more highly of the importance and value of the information contained within them and had a keener ability to understand that information. Accordingly, public health interventions contributing to strategies motivating consumers to use food labels can provide useful insight for developing promotional campaigns. This is a critical gap that needs to be addressed, as better-educated participants were more likely to understand food labels but not more likely to use labels [8,21].

The results of this study show that factors associated with the comprehensive use of food labels in Jordan are similar to those in other populations [8,23,24,25,26]. For example, Nutrition Facts used among National Health and Nutrition Examination Survey (NHANES) 2005–2006 participants were associated with sociodemographic factors such as being female, white, having high education and income, being older, and living alone. Greater use of food labels was associated with better dietary patterns, including lower sugar, total and saturated fat, as well as energy consumption [24]. NHANES 2007–2010 further reported that Nutrition Facts use was related to active engagement in weight loss activities such as physical activity and using commercial diets [23,25]. Among younger adults, female sex, higher education and income, regular use of pre-prepared meals, physical activity, overweight, and weight watching were indicators of higher use of food labels [23,25]. Similar results were also reported from the region [8,11,18,26]. For example, Shahrabani et al. [8] studied food label use among Israeli consumers and reported more frequent use by those with higher education, Arabs compared to Jews, and those who live in regions other than Tel Aviv and the center. They also found that food label use was more frequent among those who had positive Health Model Belief constructs related to food labels, such as higher levels of perceived benefit and higher confidence in reading and understanding food labels. A study in Saudi Arabia [11] evaluated consumers’ knowledge, awareness, and practices in regard to food labels and explored the effect of sociodemographic characteristics on various aspects of food labels. They reported that most participants had moderate knowledge about general nutrition, with about 57.6% of participants reading food labels, and that the relationship between knowledge and practice was significant. In Riyadh City [18], a study assessed consumers’ knowledge, attitudes, and practices toward menu calorie labeling. They reported that gender and educational attainment significantly affected consumers’ knowledge regarding calorie labeling use. In Lebanon [26], the overall mean score for knowledge, attitude, and practice related to nutritional labels was reported to be low. Gender, age, and educational level were predictors of such scores. Accordingly, the frequency of food label use seems to be globally uniform, and little variations seem to exist in terms of culture, race, or nationality. This attests to the need for regional and global actions to increase attention to the use of food labels.

The current results indicate that knowledge regarding food labels was not optimal. This may highlight a public health threat to combating NCDs and may direct attention to the intelligibility of food labels in Jordan. The traditional food labels used in Jordan may hinder effective information sharing, making them difficult to use properly. For Jordan to effectively combat NCDs, policies should consider improving information sharing from food labels. Certain food label designs are better suited to share information regarding nutritional values and to support consumers’ judgment regarding healthier diets [27]. Front-of-pack nutrition labels, including multiple traffic lights, reference intake, health star rating, nutria-score, or SENS [28] should be urgently considered in Jordan. Such labels were found to improve the nutritional quality among consumers, with evidence of decreasing the amount of energy and fats consumed and increasing fiber, fruit, and vegetable consumption [27,29,30]. These changes were secured by facilitating a better understanding and comparing the nutritional quality of foods and beverages relative to no label [27]. It is worth mentioning that a study in Saudi Arabia reported that “the tiny size and font of print and shortage of time, as well as differences in language” were barriers to understanding food labels [11]. Accordingly, a more comprehensive study of food labeling should be carried out to frame and improve information sharing that considers potential demographic differences in the population, including age, sex, and educational background. This includes assessing potential language barriers and addressing them. For example, designing software applications to translate information for consumers could be a critical need to effectively transfer information from food labels to consumers.

Given the burden of NCDs in Jordan [2] and the significant impact that food labeling programs can have on helping consumers make healthier choices [31], efforts should be made to promote increased use of food labels and to improve attitudes towards healthy diet. The results show that frequent, comprehensive use of food labels is closely associated with attitudes and views towards the perceived importance and benefit of using food labels, as well as the knowledge and ability to work through the sometimes apparently complicated information contained within food labels that may require high levels of literacy and mathematical skill. Previous research proposes proper utilization of easy-to-interpret labeling among vulnerable groups [32]. Educational campaigns addressing the use of food labels are recommended as an integral part of combating diet-related NCDs in Jordan [33].

### Limitations

While the results hold significant public health recommendations, there are also limitations worth mentioning. First, the sample may have limited generalizability given the sampling strategy. A convenient sample from conventional supermarkets may not be generalizable to the reference population, which also includes customers of other venues such as small supermarkets and convenience stores. Future research should consider understanding the attributes under investigation using consumers from convenience stores and small shops. Second, practice and attitude questions do not reflect standardized methods, and comparison with other studies needs careful consideration. Still, a standardized tool to assess knowledge, attitude, and practice related to food label use does not exist. Knowledge questions reflect familiarity with information available on a provided food label and may reflect a good measure for that specific food label. As well future studies could consider using questions (items) utilized in this study to assess levels of attitude and practice. Future studies should also consider assessing or controlling for additional factors, such as health status or dietary habits, to achieve more accurate results. The final limitation of the current study is the inability to assess differences by residence (urban vs. rural). While we believe that this was captured in education and income variables, we still believe rural and urban differences should be studied explicitly.

## 5. Conclusions

To conclude, future public health campaigns in Jordan should focus on promoting awareness of food labels, the importance and benefit of using food labels, and education on how to use the information offered within them. Strategies that can make finding and using food labels easier, such as front-of-pack labeling, should be the focus of future research. National policies to promote the implementation of simplified nutrition information on the front of food packages should be recommended in Jordan as one of the cost-effective interventions to promote a healthy diet.

## Figures and Tables

**Figure 1 nutrients-15-04893-f001:**
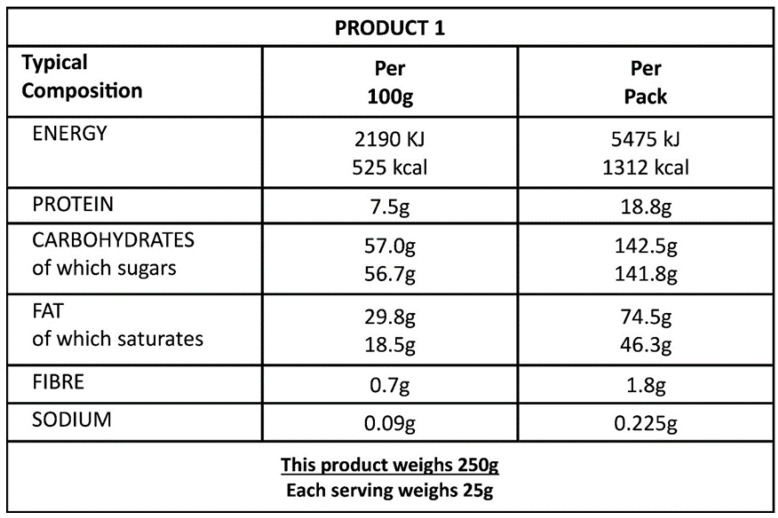
Example of a food label.

**Figure 2 nutrients-15-04893-f002:**
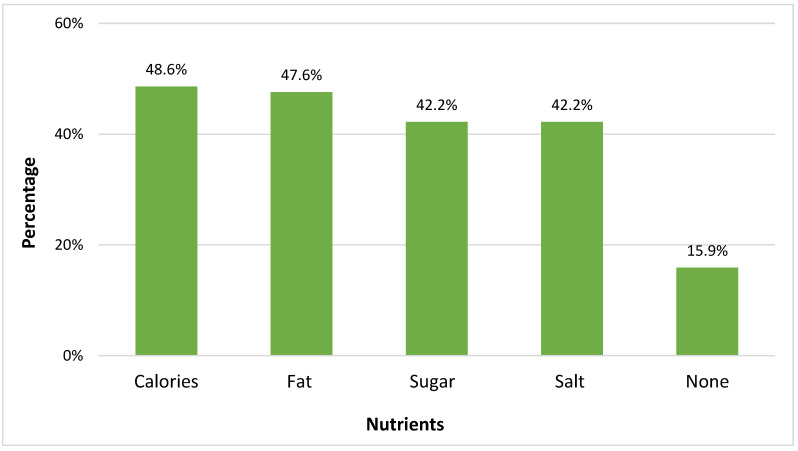
Nutritional information that consumers find important (N = 939).

**Table 1 nutrients-15-04893-t001:** Questions used to define the knowledge score (N = 939).

Knowledge Score	Correct Answer	Incorrect Answer
Questions *	n (%)	n (%)
Q1: How many grams of sugar are in two servings of the product?	239 (25.5)	700 (74.5)
Q2: How many grams of fat are in half a pack (50%)?	403 (42.9)	536 (57.1)
Q3: How many servings are in this product?	373 (39.7)	566 (60.3)
Q4: This product is low, moderate, or high in fat?	290 (30.9)	649 (69.1)
Mean knowledge (SD)	0.35 (0.33)	
Mean of total correct answers (SD)	1.39 (1.33)	

* Participants were given an example photo of a standard food label and were asked questions testing their ability to comprehend the information within it. Each question had four or five answer options with one correct answer. For each question, participants were given a score of one if answered correctly and zero otherwise. Exploratory factor analysis was subsequently used to identify questions that defined the knowledge factor.

**Table 2 nutrients-15-04893-t002:** Questions used to define the overall food label practice scale and subscales (N = 939).

Question *	Mean (SD) **
Product Information Subscale *	
In the past six months, how many times have you read the product name on the label?	3.82 (1.07)
In the past six months, how many times have you read the country of origin on the label?	3.98 (1.01)
In the past six months, how many times have you read the “best used by” a specific date on the label?	4.16 (1.00)
In the past six months, how many times have you read “best used until a specific date” on the label?	4.09 (1.06)
In the past six months, how many times have you read the preparation method on the label?	3.49 (1.12)
In the past six months, how many times have you read the storage method on the label?	3.62 (1.16)
Nutritional facts subscale *	
In the past six months, how many times have you read the serving information on the label?	3.13 (1.18)
In the past six months, how many times have you read the product name/food item on the label?	3.78 (1.11)
In the past six months, how many times have you read the nutritional claims (like low fat, heart-healthy) on the label?	3.31 (1.15)
In the past six months, how many times have you read the nutritional information on the label?	3.50 (1.06)
In the past six months, how many times have you read allergy information (like nut-free) on the label?	3.10 (1.22)
In the past six months, how many times have you read information about the ingredients (ingredients, quantity of ingredients) on the label?	3.52 (1.06)
In the past six months, how many times have you read information on the label about ingredients if they are genetically modified?	2.99 (1.30)
In the past six months, how many times have you read information on the label about ingredients if they are organic?	3.00 (1.20)
Overall food label practice score (SD)	3.54 (0.81)
Overall food label practice sum of scores (SD)	49.5 (11.36)

* Exploratory factor analysis was used to identify questions defining the food label practice scale and two subscales (product info and nutritional facts). ** Each question was scored according to a 5-point Likert scale from 1 (always) to 5 (never).

**Table 3 nutrients-15-04893-t003:** Questions used to define the overall attitude scale and subscales (N = 939).

Question *	Mean (SD) **
Label information subscale	
I frequently read the food labeling.	3.72 (0.94)
I believe food labels are important.	4.49 (0.68)
I believe food labels provide sufficient information.	3.74 (1.21)
I find the information on food labels understandable.	3.61 (0.87)
I am confident in my understanding of food labels.	3.68 (1.04)
Nutritional values subscale	
Reading the type and percentage of fat on food labels is important to me.	3.26 (1.12)
Knowing the type of added sugar in products matters to me.	3.62 (1.04)
I find it important to check if a fat product is hydrogenated when reading food labels.	3.58 (1.05)
Overall food label attitude score (SD)	3.71 (0.65)
Overall food label attitude sum of scores (SD)	29.70 (5.23)

* Exploratory factor analysis was used to identify questions defining the attitude scale as well as the label information and nutritional values subscales. ** Each question was scored according to a 5-point Likert scale.

**Table 4 nutrients-15-04893-t004:** Sociodemographic characteristics of Jordanian adult consumers (N = 939).

Sociodemographic Characteristics	
Characteristic	n (%)
Gender	
Female	398 (42.4)
Male	541 (57.6)
Age-groups	
20–30	200 (21.3)
30–40	287 (30.6)
>40	452 (48.1)
Education	
High School or less	74 (7.9)
Undergraduate	633 (67.4)
Graduate/Postgraduate	232 (24.7)
Monthly income	
200–500	388 (41.3)
500–800	205 (21.8)
800–1200	158 (16.8)
>1200	188 (20.0)
Responsible for household grocery shopping	
Yes	666 (70.9)
No	273 (29.1)

**Table 5 nutrients-15-04893-t005:** Summary of practice and attitude scale and knowledge score.

Scale/Score	Number of Items (Questions)	Minimum Value	Maximum Value	Mean ± SD	Percent of Maximum
Practice scale	14	5	70	49.5 ± 11.36	70.7%
Attitude scale	8	5	40	29.70 ± 5.23	74.3%
Knowledge score	4	0	4	1.39 ± 1.33	34.8%

**Table 6 nutrients-15-04893-t006:** Characteristics of comprehensive and non-comprehensive users of food labels (N = 939).

	Users		
Characteristic	Comprehensive	Non-Comprehensive	*p*-Value *
Overall	248 (26.4%)	691 (73.60%)	
Gender			0.011
Female	122 (30.7)	276 (69.3)	
Male	126 (23.3)	415 (76.7)	
Age-groups			0.126
20–30	46 (23.0)	154 (77.0)	
30–40	69 (24.0)	218 (76.0)	
>40	133 (29.4)	319 (70.6)	
Education			0.029
High School or less	15 (20.3)	59 (79.7)	
Undergraduate	157 (24.8)	476 (75.2)	
Graduate/Postgraduate	76 (32.8)	156 (67.2)	
Monthly income			0.902
200–500	103 (26.5)	285 (73.5)	
500–800	57 (27.8)	148 (72.2)	
800–1200	42 (26.6)	116 (73.4)	
>1200	46 (24.5)	142 (75.5)	
Responsible for shopping			<0.001
Yes	206 (30.9)	460 (69.1)	
No	42 (15.4)	231 (84.6)	
Total attitude score, mean (SD)	33.63(4.00)	28.30 (4.89)	<0.001
Total knowledge score, mean (SD)	1.61(1.36)	1.31 (1.31)	0.002

* The Chi-square test was used to compare percentages, and the student’s *t*-test was used for continuous variables.

**Table 7 nutrients-15-04893-t007:** Factors associated with comprehensive food label use among study participants.

Characteristic	Unadjusted Effect		Adjusted Effect *	
	Crude OR (95% CI)	*p*-Value	Adjusted OR (95% CI)	*p*-Value
Gender				
Female	1.46 (1.09–1.95)	0.012	1.75 (1.24–2.48)	0.002
Male	Reference		Reference	
Age-groups (years)				
20–30	Reference	Reference	--	--
30–40	1.06 (0.69–1.62)	0.790	--	
>40	1.4 (0.95–2.05)	0.091	--	
Education				
School	Reference	Reference	Reference	Reference
Undergraduate	1.3 (0.72–2.35)	0.391	--	
Graduate/Postgraduate	1.92 (1.02–3.6)	0.043	1.46 (1.00–2.13)	0.052
Monthly income (JOD)				
200–500	Reference	Reference	--	
500–800	1.07 (0.73–1.56)	0.743	--	
800–1200	1 (0.66–1.52)	0.993	--	
>1200	0.9 (0.6–1.34)	0.593	--	
Responsible for shopping				
Yes	2.46 (1.71–3.56)	<0.001	2.47 (1.63–3.74)	<0.001
No	Reference	Reference	Reference	Reference
Attitude score, mean (SD)	1.31 (1.25–1.36)	<0.001	1.39 (1.30–1.49)	<0.001
Knowledge score, mean (SD)	1.18 (1.06–1.32)	0.002	3.81 (1.37–10.57)	0.010

* The interaction term between attitude and knowledge scores was statistically significant (OR 0.96; 95% CI 0.93–0.99); the backward selection method was used.

## Data Availability

The data presented in this study are available on request from the corresponding author. The data are not publicly available because the IRB at King Abdullah University Hospital requests data availability to be approved with reasonable requests submitted through the corresponding author and the IRB committee.

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
