# Peer review of "Factors Contributing to the Comprehensive Use of Food Labels in Jordan"

_nutrients, 2023, doi:10.3390/nu15234893_

Round 1

Reviewer 1 Report

Comments and Suggestions for Authors

This study investigated the title " Factors contributing to the comprehensive use of food labels in Jordan" After revision, I think it is an interesting article. 

1. Please clarify how the minimum sample size estimate of 377 participants was calculated.

2. Building on the previous point, with a total of 939 adults participating in this study, does this deviate from the initially designed sample size?

3. Although the authors have listed the limitations, the following points still warrant attention:

   (1) Depth of discussion: Certain aspects, such as the design of food labels or language barriers, may require a more in-depth exploration. This would aid in a more comprehensive understanding of how consumers interpret and utilize food labels.

   (2) Additional control variables: When analyzing factors related to food label usage, it might be beneficial to consider more control variables, such as health status or dietary habits, to achieve more accurate results.

4. It's suggested to change the description to the third person and avoid using terms like "our" or "we" in the Discussion section.

Author Response

Thank you for your valuable notes. We have addressed them as attached.

Reviewer 2 Report

Comments and Suggestions for Authors

Manuscript ID: nutrients-2679556

Title: Factors contributing to the comprehensive use of food labels in Jordan

The study presents a cross-sectional, online survey investigating food labels as informational tools with potential to contribute to the prevention of diet-related non-communicable diseases. It describes knowledge, attitudes and practices of Jordanian consumers' as well as their relationships. Many important variables are studied to assert a conclusion and perceive the possible future interventions.

Comments:

Lines 90-91: description of the study participants includes clear separation on residents of rural and urban areas. This is a major factor that could affect the consumers¢ behavior. However, the influence of this difference was not mentioned in the results and discussion. If the results showed no grounds for the separation of these two categories of residents, that should be mentioned.

Line 93 – Material and methods: The relationship between the number of questions and scores is not clearly explaned in all segments and needs more clarification in order to allow readers to understand the total scores.

Table 2: in the subsection „ Nutritional facts subscale“ the third question is related to nutritional claims. However, given examples present a nutritional claim („low fat“) and a health claim („heart-healty“) – the authors should deal with this issue. There is general lack of discussion related to health claims, in fact, these type of claims are not mentioned at all, despite their importance and potentially strong influence on consumers' choices.

Line 190-191: delete the text from the journal template.

Lines 217-219: complete repetition of methodological information given in lines 161-164. Please delete the repeted sentences. A short comment related to the criterion used to distinguis comprehensive and non-comprehensive users could be inserted as a part of the following sentence („The number of comprehensive users“ (total practice scor equal to or greater than the 75th percentile, „was....“).

Table 6: table 6 includes the entire table 4 (the first and the second column, apart from the last two rows). Additionally, the newly presented data include percentage shares along with the absolute numbers of users – therefore,it would be more appropriate to delete repeated table 4 data.

Lines 277-291: It is important to put the current study in the context of similar investigations. In that sense, the discussion section should be extended in order to highlight similarities and differences with other studies, especially recent ones and from the same world region. For example (but not limited to), a recent, very similar but more comprehensive study from Lebanon has only be mentioned (reference number 26) under the umbrella Similar results were also reported from the region (8,11,18,26)“.

Lines 298-300 (in general): are eco-friendly labels used in Jordan? Could such labelling be of importance and interest for Jordanian consumers?

Line 316: Some additional aspects that have not been included in the study have to be mentioned and discussed in terms of their potential influence on the study results.

For example, employment profession (medical or no medical – influence on the awareness of the importance of nutritional information), existence and type of medical condition (influence on the medical advices and recommendations for the patients with diet-related health conditions and allergies), diet type (related or not to existing health condition, motivated with interest in prevention of diet-related diseases, motivated with concerns related to food safety or environmental impact (eg., organic food), element of healthy-life styles, etc). Health claims (already mentioned in comment related to table 2) should also be further discussed. Additionally, appart from the mentioned food ingredients and nutritional facts, food labels offer information on food additives – these intentionally added compounds can also be a subject of debate and affect the consumers choice. If that is not a case in Jordan, it should be mentioned as such. Furthermore, the fact that shoppers are often seeking price over quality can also have deep impact on their interest in food labels.

Author Response

Thank you very much for your valuable notes. While some of them were informative, we have included a comment regarding health claims and nutritional claims as outside the focus of the current study question. We are currently working on a subanalysis for subscales in a more focused approach.

Author Response

(The authors gave the same response as above.)

Round 2

Reviewer 2 Report

Comments and Suggestions for Authors

The manuscript has been revised according to the given comments.